# A Heuristic-Based Adaptive Iterated Greedy Algorithm for Lot-Streaming Hybrid Flow Shop Scheduling Problem with Consistent and Intermingled Sub-Lots

**DOI:** 10.3390/s23052808

**Published:** 2023-03-03

**Authors:** Yiling Lu, Qiuhua Tang, Quanke Pan, Lianpeng Zhao, Yingying Zhu

**Affiliations:** 1Key Laboratory of Metallurgical Equipment and Control Technology of Ministry of Education, Wuhan University of Science and Technology, Wuhan 430081, China; 2Hubei Key Laboratory of Mechanical Transmission and Manufacturing Engineering, Wuhan University of Science and Technology, Wuhan 430081, China; 3School of Mechatronic Engineering and Automation, Shanghai University, Shanghai 200444, China

**Keywords:** hybrid flow shop scheduling, consistent sub-lots, intermingle, heuristic-based, adaptive strategy

## Abstract

Owing to the different quantities and processing times of sub-lots, intermingling sub-lots with each other, instead of fixing the production sequence of sub-lots of a lot as in the existing studies, is a more practical approach to lot-streaming flow shops. Hence, a lot-streaming hybrid flow shop scheduling problem with consistent and intermingled sub-lots (LHFSP-CIS) was studied. A mixed integer linear programming (MILP) model was established, and a heuristic-based adaptive iterated greedy algorithm (HAIG) with three modifications was designed to solve the problem. Specifically, a two-layer encoding method was proposed to decouple the sub-lot-based connection. Two heuristics were embedded in the decoding process to reduce the manufacturing cycle. Based on this, a heuristic-based initialization is proposed to improve the performance of the initial solution; an adaptive local search with four specific neighborhoods and an adaptive strategy has been structured to improve the exploration and exploitation ability. Besides, an acceptance criterion of inferior solutions has been improved to promote global optimization ability. The experiment and the non-parametric Kruskal–Wallis test (*p* = 0) showed the significant advantages of HAIG in effectiveness and robustness compared with five state-of-the-art algorithms. An industrial case study verifies that intermingling sub-lots is an effective technique to enhance the utilization ratio of machines and shorten the manufacturing cycle.

## 1. Introduction

A hybrid flow shop is generally applied in chemical, textile, steel, and semiconductor manufacturing industries, etc. It includes multiple processing stages with one or more parallel machines at each stage. Hybrid flow shop scheduling aims to allocate exactly one machine at each stage for all the jobs involved [1] and determine the job sequence to be processed. A reasonable schedule for this workshop may effectively improve production efficiency, shorten the manufacturing cycle time, and balance the utilization ratio of machines. 

It should be pointed out that in many real-world scenarios, each job holds a lot of identical items. If all these items are treated as a job and the impact of lot sizing is ignored, as shown in Figure 1a, the production efficiency will be seriously decreased and the manufacturing cycle time will be extremely prolonged. Hence, the lot-streaming hybrid flow shop scheduling problem comes into existence to divide these items into smaller sub-lots, for the purpose that different sub-lots of a lot may be processed concurrently and the manufacturing cycle time can hence be shortened to a large extent. In view of the variability in sub-lots through all the stages in a hybrid flow shop, lot-streaming can be further classified into three categories: equal sub-lot, consistent sub-lot, and variable sub-lot [2]. Specifically, an equal sub-lot divides identical items into several sub-lots with the same sub-lot size and remains unchanged in the subsequent stages. A consistent sub-lot obtains sub-lots with different numbers of items in each and remains unchanged in the subsequent stages. A variable sub-lot achieves a different number of items in each sub-lot, and more importantly, at each stage, the number of sub-lots or the quantity of the items in each sub-lot may be different. Apparently, an equal sub-lot tends to be too idealistic and results in longer manufacturing cycle time; a variable sub-lot is too complicated and is hard to implement in practice. Comparatively speaking, a consistent sub-lot is moderate in difficulty and hence is the primary practice.

Besides, intermingling sub-lots of different jobs with each other is another practical decision. However, most of the literature assumes that all the sub-lots of a job should be allocated to only one machine consecutively, as illustrated in Figure 1b, so as to reduce the amount of switching from one job to another. This mainly arbitrary decision to avoid the intermingling of sub-lots of different jobs indeed incurs a price, as illustrated in Figure 1c, in terms of more idle time for the machines, less machine utilization, and a longer manufacturing cycle time. 

In light of the above, this work introduces the lot-streaming hybrid flow shop scheduling problem with consistent and intermingled sub-lots (LHFSP_CIS). For the solution procedure, since iterated greedy is a simple but powerful algorithm for solving optimization problems [3,4], a heuristic-based adaptive iterated greedy algorithm (HAIG) is proposed here to solve this problem. This work mainly presents the following two contributions. 

A mixed integer linear programming model is established to highlight the influences of intermingling sub-lots with each other with respect to production efficiency and sequence-dependent setups.A heuristic-based adaptive iterated greedy algorithm (HAIG) with three main modifications achieves a more balanced exploration and exploitation. The heuristic-based initialization globally minimizes the maximum completion time by relaxing the sequence-dependent setup time caused by intermingling. Consequently, four special neighborhood structures based on critical paths, and an adaptive strategy, are proposed to enhance the local search capability. Besides, an acceptance criterion of inferior solutions is improved to promote global optimization ability and avoid premature convergence.

This work is structured as follows. Section 2 carries out a literature review. Section 3 describes the considered problem and establishes a MILP model. Section 4 details the HAIG algorithm. An experimental study is conducted in Section 5 and followed by a conclusion in Section 6.

## 2. Literature Review

This section first reports the research status of the hybrid flow shop scheduling problem with consistent and intermingled sub-lots and then surveys the existing methods to solve the related problems.

### 2.1. Flow Shop Scheduling Problem with Consistent and Intermingled Sub-Lots

Owing to higher production efficiency and simpler production management, flow shop scheduling has attracted great attention from the industrial and academic communities. Concerning the practical application requirements, time-based optimization objectives are generally employed to solve the flow shop scheduling problems. These objectives include the minimization of the maximum completion times [5,6,7], the blocking or starvation times of machines [8], the earliness or tardiness times of jobs [9], and the total flowtime of jobs [10]. These objectives are beneficial to the machine utilization rate and production efficiency. However, if the lot of jobs is relatively large, all the items in a lot can only be processed after the completion of the previous stage. This obviously results in a higher work-in-process inventory and a less smoothened workflow, and notably prevents the above time-based objectives from reaching a better value.

A recent development to reduce the work-in-process inventory is to adopt the lot-streaming into the flow shop scheduling problems by splitting a lot into smaller sub-lots. The lot-streaming has been applied in various production scenarios, such as the distributed permutation flow shop [11], the two-stage assembly hybrid flow shop [12], the blocking flow shop [13], and the hybrid flow shop [14]. It should be noted that in these studies, all sub-lots in a lot are assumed to be serially processed by a machine for simplicity. The production sequence of these sub-lots is strictly fixed, which avoids the intermingling of a sub-lot with another lot. This non-intermingling lot-streaming technique is unreasonable due to the following two reasons. First, all sub-lots of a lot must be allocated to exactly one machine. If there are tremendous items in a lot, the maximum completion time completely equals the processing time of this lot, implying the invalidation of the lot-streaming technique. Second, machine setup time cannot be negligible in the hybrid flow shop scheduling problem considering lot-streaming [15] and serially processing all sub-lots of a lot is definitely conducive to reducing the sequence-dependent setup times. However, this strict restriction is not necessarily favorable to the maximum completion time.

Intermingling sub-lots of different lots emerged in similar industrial problems such as permutation flow shop scheduling problems. For instance, Feldmann et al. (2008) [16] introduced the sub-lot intermingling of particular products for permutation flow shop problems. Mortezaei et al. (2014) [17] studied a permutation flow shop with preventive maintenance and consistent intermingled sub-lots. As for the hybrid flow shop scheduling problems, Zhang et al. (2014) [18] introduced the intermingling of equal sub-lots into the flow shop scheduling problem. Obviously, consistent sub-lots are considered, implying the number of items in each sub-lot may be different. Thus, this work studies the lot-streaming hybrid flow shop scheduling problem with consistent and intermingled sub-lots to enhance the utilization ratio of machines and the manufacturing cycle.

### 2.2. Meta-Heuristic Algorithms

In recent decades, many algorithms have been proposed to tackle the hybrid flow shop scheduling problem and its variants. These algorithms include mathematical formulations [17,19,20,21,22], swarm optimization algorithms [11,12,18], and local search algorithms [13,23]. The first can obtain the optimal solution for small-sized cases at the cost of exponentially increased computation time [20]. To tackle medium- or large-sized cases, the latter two types of algorithms are commonly utilized. Concretely, swarm intelligent algorithms have a simple structure, and good global convergence, but are prone to premature convergence and it is difficult to find the optimal solution due to poor local optimization ability [24]. In contrast, the local search ones show stronger exploitation capacity via exquisite problem-specific neighborhood search strategies.

Thus, this work uses the local search algorithm, specifically iterated greedy algorithm (IG), a simple but powerful algorithm [3,4], to solve the HFSP-CIS. Since the local search algorithm is to search around a solution in its local scope, the performance of the initial solution and search strategies are particularly important.

To improve the performance of the initial solution, there are miscellaneous classical rules for hybrid flow shop scheduling problems, i.e., shortest processing time (SPT) and longest processing time (LPT) [25]. To be more relevant, this work customizes a heuristic-based initialization to generate the initial solution for the IG to further reduce machine idle times, which enables the generation of a high-quality initial solution.

To improve the local search ability, many operators are designed with the insertion and pairwise exchange operators [26], an exploration heuristic [27], an insert move with two job selection mechanisms [28], and a restart scheme with six operators [29]. Therefore, for the improvement of the exploitation ability, this work specifically designs four kinds of neighborhood structures and an adaptive strategy. Besides, an inferior solution improvement mechanism is attempted to improve the exploration ability of the algorithm.

Based on the above, this work studies the lot-streaming hybrid flow shop scheduling problem with consistent and intermingled sub-lots (LHFSP-CIS) and aims to minimize the maximum completion time. To tackle this problem, a heuristic-based adaptive iterated greedy algorithm (HAIG) is proposed. Specifically, three operators for initialization, local search, and acceptance criterion are dexterously modified.

## 3. Results

### 3.1. Problem Formulation

The considered problem LHFSP-CIS is described as follows. Several lots are to be processed through a set of stages with more than one parallel machine at each stage. The framework of the hybrid flow shop environment is shown in Figure 2. Each lot can be split into several sub-lots with different numbers of items. The number of sub-lots and the quantity of items in each sub-lot remains unchanged through all the stages. At each stage, all the sub-lots of a lot can be processed by different machines; or, if by a machine, sub-lots of other lots can be intermingled among them. In each sub-lot, all the items should be processed continuously and their processing time is the product of the number of items in this sub-lot and the processing time per item. Besides, sequence-dependent setup time is expected if two consecutive sub-lots are different lots and otherwise are unnecessary. The assumptions for this problem are summarized below. 

All items of all lots are available at time zero.The number of sub-lots of a lot is limited to its maximum value.At each stage, each sub-lot should be allocated to exactly one machine and all sub-lots of a lot may be allocated to more than one machine.On each machine, idle times between any two consecutive sub-lots are allowed, and all items in a sub-lot should be processed consecutively and without a break.The buffer capacity between the two stages for storing intermediate products is infinite.Each sub-lot can be transported to the next stage only after its completion at the current stage.A machine can process at most one sub-lot at a time. It can start processing only if it gets ready and the corresponding sub-lot arrives.On each machine, setup is compulsory if two consecutive sub-lots are from different lots while it is unnecessary if both of them are from the same lot.

### 3.2. Mathematical Model

The considered problem needs to determine the sub-lot splitting of all lots, the machine assignment of sub-lots at each stage, and the production sequence of sub-lots at each stage. Particularly, the sub-lot splitting includes determining the number of sub-lots of a lot and the number of items in each sub-lot. To formulate the mathematical model, the notations used are given as Table 1.

With the notations listed above, a mixed integer linear programming model for the LHFSP_CIS is formulated as follows.
(1)$$Min\;C_{max}$$
(2)$$C_{max}\geq F_{i,j,e},\forall j\in J,e\in E,i=I$$
S.T.
(3)$$W_{j,e}\geq \frac{Q_{j,e}}{\Psi },\forall j\in J,e\in E$$
(4)$$W_{j,e}\leq Q_{j,e},\forall j\in J,e\in E$$
(5)$$\sum_{e=0}^{L_{j}}Q_{j,e}=n_{j},\;\forall j\in J$$
(6)$$\sum_{k=0}^{M_{i}}D_{i,k,j,e}=1,\forall j\in J,i\in I,e\in E$$
(7)$$Y_{i,k,j,e,j\prime ,e\prime }+Y_{i,k,j\prime ,e\prime ,j,e}\leq D_{i,k,j,e},\;\forall \left(j,j\prime \right)\in J,i\in I,\left(e,e\prime \right)\in E,k\in M_{i}$$
(8)$$Y_{i,k,j,e,j\prime ,e\prime }+Y_{i,k,j\prime ,e\prime ,j,e}\leq D_{i,k,j\prime ,e\prime },\;\forall \left(j,j\prime \right)\in J,i\in I,\left(e,e\prime \right)\in E,k\in M_{i}$$
(9)$$Y_{i,k,j,e,j\prime ,e\prime }+Y_{i,k,j\prime ,e\prime ,j,e}\geq D_{i,k,j,e}+D_{i,k,j\prime ,e\prime }-1,\forall \left(j,j\prime \right)\in J,i\in I,\left(e,e\prime \right)\in E,k\in M_{i}$$
(10)$$F_{i,j,e}-B_{i,j,e}\geq p_{i,k,j}\cdot Q_{j,e}-\Psi \times \left(1-D_{i,k,j,e}\right),\forall j\in J,i\in I,e\in E,k\in M_{i}$$
(11)$$F_{i,j,e}-B_{i,j,e}\leq p_{i,k,j}\cdot Q_{j,e}+\Psi \times \left(1-D_{i,k,j,e}\right),\forall j\in J,i\in I,e\in E,k\in M_{i}$$
(12)$$B_{i,j\prime ,e\prime }-F_{i,j,e}-s_{i,j,j\prime }\geq -\Psi \times \left(3-Y_{i,k,j,e,j\prime ,e\prime }-D_{i,k,j,e}-D_{i,k,j\prime ,e\prime }\right),\forall \left(j,j\prime \right)\in J,i\in I,\left(e,e\prime \right)\in E,k\in M_{i}$$
(13)$$B_{i+1,j,e}-F_{i,j,e}-tr_{i}\cdot W_{j,e}\geq 0,\forall j\in J,i\in \left[1,\ldots ,I-1\right],e\in E$$

The objective is to minimize the maximum completion time $C_{max}\;$ as specified in Equations (1) and (2). Note that, the maximum completion time cannot be less than the finishing time of each sub-lot of each lot at the last stage.

All the constraints are divided into three categories: sub-lot splitting, production sequence, and sub-lot timing. Regarding sub-lot splitting constraints, Equations (3) and (4) state that if the quantity of items in a sub-lot is larger than 1, i.e., $Q_{j,e}\geq 1$, this sub-lot is utilized, and hence $W_{j,e}=1$. Equation (5) ensures that all items in a lot are allocated to the sub-lots related to this lot. Obviously, the number of sub-lots and the quantity of the items in each sub-lot are the same from stage to stage, and hence both of them remain unchanged through all the stages.

In respect to production sequence constraints, Equation (6) points out that each sub-lot must go through all the stages, and at each stage, it must be allocated to exactly one machine. Since the allocated machines of two sub-lots of a lot are not specifically restricted, both of them can be the same machine or two different machines. Besides, Equations (7)–(9) line up all the sub-lots allocated to a machine into a sequence. Obviously, sub-lots of a lot are not required to be consecutively processed or without a break. In other words, they can be separated and hence intermingling sub-lots of different lots can be accomplished.

To handle the sub-lot timing constraints, Equations (10) and (11) demand that the processing of each sub-lot cannot be interrupted as long as it starts. Equation (12) describes two scenarios on a machine for starting the processing of a sub-lot: (1) if this sub-lot and its previous one are from the same lot, machine setup is unnecessary and hence this sub-lot can start after the completion of the previous; (2) if this sub-lot and its previous one are from two different lots, this sub-lot can start only after the completion of machine setup. Besides, Equation (13) guarantees that each sub-lot can start only after it has been transported from the previous stage to this one.

In sum, with Equations (1)–(13), LHFSP_CIS is formulated into a mixed integer linear programming model. 

### 3.3. Complexity Analysis

The above model includes discrete variables {$W_{j,e}$,$D_{i,k,j,e}$,$\;Y_{i,k,j,e,j\prime ,e\prime }$,$\;S_{j,e}$}, continuous variables {$B_{i,j,e},F_{i,j,e}$,$C_{max}$}, and 12 constraints. The number of discrete variables is vL, fvrL, $fv^{2}L^{2}r$, vL, respectively. The number of continuous variables is fvL, fvL and 1, respectively. In conclusion, the total number of variables in this MILP model is $vL\left(3+2f+r\left(f+fvL\right)\right).$ Similarly, the total number of constraints is approximately $vL\left(5+4f+r\left(2f+2fvL\right)\right)+1$. The number of constraints grows polynomially as the number of lots grows or as the maximum number of sub-lots of each lot increases. Hence, a metaheuristic algorithm is urgently needed to effectively and efficiently solve this problem with medium- and large-sized cases.

## 4. Heuristic-Based Adaptive Iterated Greedy Algorithm

The iterated greedy algorithm (IG) [3] was proposed by Ruiz in 2007 as a simple and effective meta-heuristic algorithm, which includes four processes: initialization, destruction–construction, local search, and acceptance strategy. To deal with the unique characteristics of LHFSP-CIS, i.e., sub-lot splitting, production sequencing, and machine assigning, this section advances a set of problem-specific optimization techniques. According to the following modifications, this enhanced algorithm is coined as a heuristic-based adaptive iterated greedy algorithm (HAIG).

The procedure of HAIG is shown in Algorithm 1, where UpdateBestX replaces the incumbent best solution with a better one just obtained, and UpdateArchiveX substitutes the worst solution in the archive set as long as a better one is achieved.
**Algorithm 1** The procedure for the HAIG1: **Define** the termination criterion T2: **Initialize** the primary solution X0
←Heuristic-based Initialization()  //*initialization*3: X←Adaptive Local searchX0  //*Local search*4: UpdateBestX5: UpdateArchiveX6: **While** T is not satisfied **do**7:  X′←D/CX    // *The destruction-construction (DC) process*8:  X″←Adaptive Local searchX′9:  UpdateBestX″10:  UpdateArchiveX″11:  X←Improved acceptance criterionX″12: **End While**13: **Return**
Xbest


### 4.1. Encoding with Decoupling Strategy

The encoding encompasses two facets of a solution on sub-lot splitting and production sequencing at the first stage, and a two-layer representation of solutions is thus presented. It should be noted that the decision variable on machine assignment for each sub-lot at each stage is handled in the decoding process. 

The first layer represents the sub-lot splitting via a v∗L-dimensional matrix, $Z_{v\times L\;}=\left[Z_{1},\ldots ,Z_{j},\ldots ,Z_{v}\right].$ In this matrix, each row stands for a lot, each column represents the serial number of sub-lots, and each element implies the number of items in the corresponding sub-lot split from a given lot. Notably, the total quantity of items in all sub-lots of a lot is equal to that from the lot. Take Equation (14) as an example. For lots {1, 2, 3, 4}, there are {3, 2, 1, 2} sub-lots with the number of items {(10, 22, 31), (32, 26), (23), (40, 14)}, respectively. Hence the total quantity of items in these lots is {63, 58, 23, 54}, respectively. In particular, the sub-lots with a quantity of zero imply that these sub-lots exist in name only.
(14)$$Z_{4\times 3}=\left[\begin{array}{ccc} 10 & 22 & 31\\ 32 & 26 & 0\\ 23 & 0 & 0\\ 40 & 14 & 0 \end{array}\right]$$

The second layer concerns the production sequence of these sub-lots at the first stage via a v∗L-dimensional permutation $\;\pi_{v*L}=\left\{\pi_{11},\ldots ,\pi_{je},\ldots ,\pi_{vL}\right\}$. Among this permutation, $\pi_{je}$ represents the sub-lot e of lot j. The sub-lot appearing to be at the front in this permutation is endowed with a higher priority to be processed at the first stage. Besides, the number of elements in this permutation equals v∗L, indicating that all sub-lots including those with zero items are sequenced in a line. As illustrated in Equation (15), the first sub-lot of lot 1 starts first, then the first sub-lot of lot 2, and so on. Clearly, the second sub-lot of lot 4 (4.2) starts earlier than the first sub-lot of lot 4 (4.1). This implies that all sub-lots of a lot are not sequenced in the strictly increasing serial number of these sub-lots as in [11]. Owing to a higher degree of freedom, the intermingling of sub-lots from the same or different lots is realized.
(15)$$\pi_{12}=\left\{1.1,\;2.1,\;3.1,\;4.2,\;1.3,\;1.2,\;3.3,\;4.1,\;2.2,\;2.3,\;4.3,\;2.3\right\}$$

Clearly, Equation (14) focuses on sub-lot splitting while Equation (15) focuses on production sequencing. However, both of them are strictly consistent in terms of the maximum number of sub-lots, and, accordingly, the process of encoding the production sequencing is independent of that of the sub-lot splitting. This encoding strategy decouples the sub-lot-based connection between these two layers. The representation at each layer is entirely free and unrestricted, and hence has the potential to cover the complete solution space.

### 4.2. Decoding

To obtain a feasible solution, two further discrete variables need to be determined: a production sequence for all the sub-lots at other stages, and machine assignment for all sub-lots at all stages including the first. 

Regarding the production sequence, except for the first stage, a “first come, first processed” is employed. The objective is to endow the sub-lot whose arrival time is earlier with a higher privilege to be scheduled. Hence, the sub-lot will start as early as possible to reduce the valueless waiting time and shorten the maximum completion time. 

Meanwhile, since multiple parallel machines are involved at each stage, the heuristic rule of “first completed, first allocated” is employed. As a result, the machine with an earlier available time has a higher priority to be allocated. Here, there are two possibilities for the earliest available time for machines. First, if the two sub-lots consecutively processed by a machine are from two lots, implying sub-lots of different lots are intermingled, the earliest available time is the completion time of the previous sub-lot plus the setup time. Second, if the two sub-lots are of a lot, the earliest available time is precisely the completion time of the previous sub-lot. If all the sub-lots can be started at the earliest available time, the utilization rate of all machines will be improved and the manufacturing cycle time will be largely reduced. The detailed decoding process is shown in Algorithm 2.
**Algorithm 2** The procedure for decoding**Input**:Sub-lot splitting, Zv×L ;  production sequence of sub-lots at the first stage,πv∗L=π11,…,πje,…,πvL**Output**: A schedule.1: **For**i=1  to f
**do**2:   Initialize the time that parallel machines are ready for setup3:   **For** all sub-lots according to the production sequence πv∗L **do**4:   **For**
m=1 to Mi **do**5:    Calculate the available time of machine m after transportation and sequence-dependent setup as Equations (12) and (13)6:   **End For**7:   Assign the sub-lot πn to the machine M with the earliest available time8:   Calculate Cmax of πn according to the earliest available time and processing time9:  **End For**10:  Update production sequence πv∗L by completing time ascending11: **End For**

### 4.3. Heuristic-Based Initialization

As declared in [30], the speed of an iterative search highly depends on the quality of the initial solution. To provide an initial solution with better quality, a heuristic-based initialization rule has been designed. For the sub-lot splitting, a more-balanced sub-lot splitting strategy for the first stage is proposed to divide the total quantity of items in a lot as evenly as possible. Under this circumstance, the considered problem has been transformed into a hybrid flow shop scheduling problem with sequence-dependent setup time. Since the shortest processing time (SPT) rule shows asymptotic optimality for the flow shop problem with the objective of mean completion time [31], an improved version of the SPT rule has been designed for the generation of the production sequence. Specifically, instead of the shortest processing time, the smallest summation of processing time and sequence-dependent setup time is regarded as the evaluation criterion. The procedure for this heuristic-based initialization is shown in Algorithm 3.
**Algorithm 3** The procedure for heuristic-based initialization**Input:** Losts J; Quantity of items in lots nj; Maximum number of sub-lots in each lot L**Output:** an initial solution X0=〈πv∗L0, Zv×L0〉1: **Define** a set ΩU that accommodate the unscheduled sub-lots2: **Define** a set STU that accommodate the summation of processing time and sequence-dependent setup time3: **For**
j=1
**to**
v
**do**      // More-balanced sub-lot splitting4:  **For**
e=1
**to**
L−1 **do**5:    Zv×L0=⌊nj/L⌋6:  **End For**7:  Zv×L0=nj−L−1×⌊njL⌋8: **End For**9: **For**
n=1 to v∗L
**do  //** Improved SPT-based production sequencing10:  STU=ϕ11:  **For**
m=1
**to**
$M_{1}$
**do**12:    **For**
p=1
**to**
ΩU
**do**13:     Calculate STmΩpU14:     STU←ST15:    **End For**16:  **End For**17:   πv∗L0←minmϵM1,pϵΩUSTU18: **End For**

### 4.4. Destruction–Construction Phase 

In the traditional iterative greedy algorithm, a fixed number of elements need to be removed in the destruction phase and subsequently inserted into all the possible positions in the remaining sequence in the construction phase. Obviously, as the size of instances increases, the computational cost for inserting the removed elements back into all the possible positions increases. To enhance the effectiveness and efficiency of this destruction and construction, a constant S representing the proportion of the number of sub-lots being destroyed to that of all sub-lots is introduced. For this, the number of elements being destroyed increases as the size of the problem instances grows. Moreover, another constant R is introduced to determine the proportion of the number of insertion positions to that of all the possible positions for construction. The procedure for this destruction and construction process is shown in Algorithm 4.
**Algorithm 4** The procedure for the destruction-construction (DC) process**Input:** an initial solution X0=〈πv∗L0, Zv×L0〉**Output:** a novel solution X1=〈πv∗L1, Zv×L0〉1: **For** i=1
*to*
v×L×S **do**    // Destruction process2:  Randomly remove sub-lot πk from π03:  πR←πk4: **End For**5: **For**i=1
*to*
v×L×S **do**    // Construction process6:  **For**
j=1
*to*
 R×all possible positions of π0 **do**7:    π1← Insert πRi into a random position of π08:    Select the best π19:    π0=π110:  **End For**11: **End For**

### 4.5. Adaptive Local Search

Neighborhood structures play a determinant role in the exploration and exploitation within the solution space [32]. To assure the local search is in a good direction, the critical path was investigated. Besides, to make the most of the neighborhood structures and avoid the waste of precious computational resources, an adaptive strategy was designed. 

#### 4.5.1. Critical Path-Based Neighborhood Operators

For LHFSP-CIS, the critical path is the longest path from the first processed sub-lot at the first stage to the last sub-lot at the last stage without any idle time. The length of the critical path is the maximum completion time and thus any movement in shortening the critical path may be conducive to the resolution of the considered problem. Hence, four types of neighborhood structures based on the critical path, as shown in Figure 3, were designed to enhance the exploitation ability in a neighborhood search, which are straightforwardly abbreviated as sub-lot splitting and three combined structures which respectively combine insertion, swap, and shuffling with sub-lot splitting. Among them, the first is designed only for the modification of the sub-lot splitting variable, and the rest of them are for the sub-lot sequencing variable. 

The first neighborhood works for the mutation of sub-lot splitting. Specifically, the sub-lots located on the critical path are critical sub-lots. Among them, the sub-lot for which the corresponding machine spends the largest waiting time is the one most promising to be improved (abbreviated as the most promising critical sub-lot). Hence, the number of items in this sub-lot was randomly reduced until the number of items in this sub-lot is greater than zero. Accordingly, the reduced items were appended to two other sub-lots which also belong to the lot of the most promising critical sub-lot. It should be noted that the quantity of items appended to these two sub-lots is also randomly determined. 

As for the rest of the three neighborhoods, sub-lot insertion directly dispatches the most promising critical sub-lot to one of the other parallel machines which are not on the critical path. Sub-lot swap interchanges the most promising critical sub-lot with the one that belongs to the same lot and has the smallest quantity of items. Sub-lot shuffling records the current production sequence, sequentially takes out all the sub-lots on the critical path, interchanges all of them randomly, and reinserts them back into the sequence. To avoid overlapping with the destruction–construction effect, the mutation of sub-lot splitting is turned on after the neighborhood operation for the processing sequence is completed.

#### 4.5.2. Adaptive Strategy

To explore a larger solution space and promote computational efficiency, an adaptive strategy is proposed to control the threshold of the maximum consecutive iteration time in a local search cycle. Specifically, all of the above four neighborhoods are implemented one by one in a fixed order. A new local search cycle starts either with the current neighborhood as long as the incumbent solution is updated, or with the next neighborhood if the corresponding threshold is reached. Obviously, as demonstrated by Equations (16)–(18), these neighborhoods are endowed with equal thresholds as the algorithm starts. As the algorithm proceeds, these thresholds continue to decrease. If a neighborhood always fails to update the incumbent solution, most computational efforts are futile and hence the corresponding threshold is largely decreased to a relatively smaller number. For this, the computational efficiency is significantly improved.
(16)$$C_{k,t}=C*exp\left(-x\;/\;0.5*\beta_{k}\right)$$
(17)x=t/T
(18)$$\beta =d_{k}/D_{k}$$
where $C_{k,t}\;$ is the threshold of the maximum consecutive iteration time for the neighborhood k at the CPU time t;  C is the initial value of $C_{k,t}$; t is the already used CPU time in the running process; T is the maximum CPU time; $d_{k}$ is the accumulated times for updating the incumbent solution by the neighborhood k; $D_{k}$ is the total times for running a local search cycle with the neighborhood k. 

The procedure of a local search with four neighborhoods and adaptive strategy is detailed in Algorithm 5.
**Algorithm 5** The procedure for local search**Input:** a solution X0, C0, Dk, dk, T, t   a set of defined neighborhoods NK,k=1,…,kmax**Output:** the updated solution X01: k←12: **While**
k<kmax
**do**3:   Dk←Dk+14:   Ck,t=C∗exp−x / 0.5∗βk5:   **While**failure<Ck
**do**6:        X1←NeighborhoodX0,NK7:        If fX1<fX0 **then**8:         X0←X19:         failure←110:         dk← dk+111:         Dk←Dk+112:         x=t/T13:         βk=dk/Dk14:         Ck,t=C∗exp−x / 0.5∗βk15:        **Else**16:         failure←failure+117:        **End If**18:     **End While**19:     k←k+120: **End While**

### 4.6. An Improved Acceptance Criterion 

To promote global optimization ability and avoid premature convergence, a certain number of inferior solutions should be accepted. However, the accepted inferior solution may drag down the search process, or lead in the wrong direction. Thus, an improved acceptance criterion is proposed to provide a corrected inferior solution. 

First, the Metropolis acceptance criterion was employed to accept the inferior solution within a certain range. When the new solution is worse than the incumbent one, the time-dependent acceptance index is calculated with Equation (19). As long as this acceptance index is greater than the newly generated random number, the inferior solution is accepted. Obviously, as the algorithm proceeds, the probability of accepting the inferior solution decreases. Hence, the algorithm tends to explore a larger solution space at the earlier stage and focuses more on the local convergence at the later one. A better tradeoff between exploration and exploitation is thereafter achieved.
(19)$$Y=exp\left(-\left(X-X_{0}\right)\;/\;T\right),\;\;\;\;\;\;\;X>X_{0}$$
where Y is the time-dependent acceptance index; X is the maximum completion time of the new solution; $X_{0}$ is the maximum completion time of the incumbent solution.

Subsequently, the accepted inferior solution is corrected from two layers of production sequencing and sub-lot splitting successively. As shown in Figure 4, for the production sequencing, the order crossover operator is utilized to randomly truncate a partial segment of sub-lots from the accepted inferior solution, and then append other sub-lots to the end of this segment in the order in another solution randomly selected in the archive set. For the sub-lot splitting, a mutation operator is designed to change the size of the sub-lots. Concretely, the size of a randomly selected sub-lot in the accepted inferior solution is reduced and the reduced is appended to one of the other sub-lots belonging to the same lot. Regarding whether to accept the corrected inferior solution, the number of consecutive failures in comparing the corrected and incumbent inferior solutions is recorded. If this failure number achieves the maximum limit, this correction process is terminated. 

The procedure of this improved acceptance criterion is in Algorithm 6.
**Algorithm 6** The procedure of the improved acceptance criterion**Input:** the current solution X0=〈πv∗L0, Zv×L0〉, the updated solution X1=〈πv∗L1, Zv×L1〉**Output:** the novel current solution X0=〈πv∗L0, Zv×L0〉1: **If**
CmaxX1<CmaxX0 **then**    // *Acceptance criterion*2: X0=X13: Xbest← UpdateBestX04: UpdateArchiveX05: **Elseif**
(random≤exp−Cmaxπ‴−Cmaxπ/Temperature) **then**6: X0=X17: UpdateArchiveX08: **While**
count<A
**do        //** *Optimal operator*9:    πv∗Lnovel←SPπv∗L010:    Zv×Lnovel←SSM Zv×L011:    **If**
CmaxXnovel<CmaxX0 **then**12:     count=013:      X0=Xnovel14:    **else**15:     count←count+116:      **End If**17:    **End While**18: **End If**

## 5. Experiment Results and Analysis

To verify the effectiveness of the proposed HAIG, four groups of experiments were carried out. After parameter calibration for all the involved algorithms, the performance of heuristic-based initialization was first tested. Then, the effectiveness of improved operators, which include the critical path-based neighborhood, the adaptive strategy, and the improved acceptance criterion were evaluated. Subsequently, the proposed HAIG was validated through a comparison experiment with state-of-the-art algorithms and a real-life industrial case.

HAIG and all the comparison meta-heuristics were programmed using C++ on Visual Studio 2019 and run on a computer with Intel(R) Core (TM) i7-10700 CPU, 2.90 GHz, 16.0 GB. All experiments used the same termination criterion, CPU time = v×L×f×M×t milliseconds. Here, t was set as 3. By using this criterion, the termination criterion is related to the total number of sub-lots, the number of stages, and the number of machines, and hence the computational time may be prolonged modestly as the problem size increases. 

### 5.1. Experimental Setting 

There were 36 examples set up for the verification experiments. Each case was represented by the number of lots, stages, and the type of machine configuration [33,34]: n×i×m. Specifically, the lot number n has nine levels {4, 12, 20, 36, 46, 56, 70, 85, 100}, and the stage number i has two levels, {3, 5}. All machines are laid out in two configurations: (a) one machine at the first stage and three machines in the rest stages; (b) three machines for each stage. Therefore, the total number of instances was  9×2×2=36, among which the number of small, medium, and large-scale cases were 12, respectively, according to the lot number.

Other parameters in the problem were set based on the real-life industrial case in the last section. Specifically, the number of items for each lot is uniformly distributed in U [10, 60], the sequence-dependent setup time and transportation time in U [10, 30] and U [20, 30], respectively, and the processing time per item in U [1, 10]. Besides, the maximum number of sub-lots of a lot was set to 3.

To simplify the algorithmic evaluation, the relative percentage deviation (RPD) between the maximum completion time calculated by the current algorithm $c_{current}\;$ and the best result of all algorithms $c_{Best}$ was selected as the performance evaluation index.
(20)$$RPD=\frac{\left(c_{current}-c_{Best}\right)\;}{c_{Best}}\times 100$$

### 5.2. Parameter Calibration

To eliminate the influence of parameters among algorithms, Design of Experiments (DOE) was utilized to set the algorithm parameters appropriately. HAIG involves five parameters: the number of solutions in the archive set (P), the initial threshold in adaptive local search (C), the number of consecutive failures in the improved acceptance criterion (A), the proportion of the number of sub-lots being destructed to that of all sub-lots (S), and the proportion of the number of insertion positions for construction (R). The Taguchi experiment [35] was used to make an empirical study on the influence of these five parameters. 

For each parameter, four reasonable levels were determined in preliminary experiments, which were P (10, 20, 30, 40); C (5, 10, 15, 20); A (5, 10, 15, 20); S (0.2, 0.3, 0.4, 0.5); R (0.2, 0.4, 0.6, 0.8). Their combinations were determined by the orthogonal array $L_{16}$, which was tested on 9 cases, namely 4 × 3_a, 12 × 3_a, 12 × 3_a, 20 × 3_a, 46 × 3_a, 56 × 3_a, 70 × 3_a, 85 × 3_a, and 100 × 3_a. Each combination was run 10 times, and the average overall RPD value was taken as experimental results, as shown in Figure 5, in which the mean signal-to-noise ratio (SNR) was utilized as the evaluation index. 

Obviously, the best combination of parameters for HAIG is {P = 20, C = 30, A = 20, S = 0.2, R = 0.7}. Therefore, these values were applied in the following experiments. Other algorithms for Improved Genetic Algorithm (IGA), Discrete Particle Swarm Optimization (DPSO), Discrete Whale Swarm Algorithm (DWSA), Discrete Grey Wolf Optimizer (DGWO), and Tabu-search optimization (TS) in this work were similarly tested and the best parameter combinations for them are in Table 2.

### 5.3. Effectiveness of Heuristic-Based Initialization 

Four initialization rules, including the random (Ran), shortest processing time (SPT), Nawaz–Enscore–Ham [36] (NEH), and collaborative rule [14] (Col), were employed for comparison experiments. For fairness, all operators in the tested algorithms are the same except for the initialization operator. All 36 cases were tested. With min RPD and mean RPD as the response value, the response value of the 95% confidence intervals is reported in Figure 6 and the detailed results are in Table 3. 

Clearly, by using the heuristic-based initialization (HBI), the obtained min and mean RPD outperform the other comparison counterparts in most instances. Further analysis reveals that the integration of sequence-dependent setup times into the initialization process balances the maximum completion time and setup times, and makes the heuristic-based initialization better than SPT and the Collaborative. Besides, accurately generating production sequences based on problem-specific rules can make the performance of the initial solutions good and stable, and certainly better than randomization in NEH. As a result, heuristic-based initialization is superior to traditional heuristic rules.

### 5.4. Effectiveness of Three Improvements

Three improved improvement operators, which include the critical path-based neighborhood operators, the adaptive strategy, and the improved acceptance criterion are verified in this section. For this, three comparison algorithms, ${\mathrm{HAIG}}_{\mathrm{rcr}},{\;\mathrm{HAIG}}_{\mathrm{rad},}{\;\mathrm{HAIG}}_{\mathrm{rim}}$ are proposed. Concretely, based on the proposed HAIG, ${\mathrm{HAIG}}_{\mathrm{rcr}}$ removes the critical path-based neighborhood operator, ${\mathrm{HAIG}}_{\mathrm{rad}}$ removes the adaptive strategy, and ${\mathrm{HAIG}}_{\mathrm{rim}}$ removes the improved acceptance criterion. Similarly, all algorithms were tested 10 times on 36 cases. The results are reported in Figure 7.

Each of the three operators shows advantages in improving the performance of HAIG in terms of the min and mean RPD. Particularly, the improved acceptance criterion plays the most important role. Besides, the marked improvement of HAIG demonstrates that these three operators are complementary and hence the combination of these three operators is effective for the proposed algorithm. 

### 5.5. Effectiveness of the Proposed HAIG

To further analyze the comprehensive performance of the proposed HAIG, five state-of-the-art algorithms were employed for comparison, which included four swarm intelligent algorithms, and one well-known local search algorithm. The four swarm intelligence algorithms were IGA [37], DPSO [38], DWSA [39], and DGWO [40]. The local search algorithm was TS [41]. For fairness, all the algorithms adopted the encoding and decoding strategies proposed in this work and the involved parameters are calibrated in Section 5.2. The best (Best) and average (Avg) maximum completion time obtained across 10 independent runs are reported in Table 4 and the plots for RPD of all comparison algorithms are depicted in Figure 8. 

Among the 36 cases, all the best and 35 average values of HAIG were better than those by the other comparison algorithms. From the statistical results, the proposed algorithm was significantly superior in terms of best solution and robustness, as confirmed in Figure 8. Above all, the significant effect of the proposed HAIG can be demonstrated. The non-parametric Kruskal–Wallis test was used to obtain *p* = 0, which is far less than 0.05, demonstrating that the performance of the proposed algorithm outperforms others. However, the proposed algorithm is only validated for this problem and cannot guarantee its performance for other problems.

### 5.6. An Industrial Case

A realistic industrial case in passenger vehicle manufacturing is taken as an example. The manufacturing process is divided into three stages: welding, painting, and assembly. Different carriages (lots) are processed through these stages one by one, and each stage has several parallel lines. The transportation time between two successive stages is given. When different types of carriages are processed in sequence, sequence-dependent setup time is involved. The case data are shown in Table 5a,b.

Results show that the best production sequence at the first stage is {1-3,4-1,1-2,4-2,1-1,2-3,4-3,2-1,3-1,3-3,2-2,3-2}, and the best sub-lot splitting is {(16, 6, 12), (8, 8, 14), (17, 11, 12), (6, 8, 6)}. Besides, the maximum completion time obtained is 389. The scheduling scheme is shown in Figure 9.

Obviously, compared with the traditional research in which all sub-lots of a lot are assigned to exactly one machine at each stage and performed serially, the proposed approach with intermingling sub-lots increases the sequence-dependent idle times to a certain extent. However, it achieves two significant advantages. First, all sub-lots are completed at the last stage almost at the same time, which increases the utilization rate of machines. Second, owing to intermingling, sub-lots of a lot can be processed on more than one machine at the same time, which shortens the manufacturing cycle time of a lot and reduces the inventory.

## 6. Conclusions and Future Work

Owing to the advantages of higher machine utilization and shorter manufacturing cycle time, intermingling sub-lots of one lot with another is a practical implementation. For this practical lot-streaming hybrid flow shop scheduling problem with consistent and intermingled sub-lots, this work established a mixed integer linear programming (MILP) model and proposes a heuristic-based adaptive iterated greedy algorithm (HAIG) with three modifications to solve this problem. The result (335) was obtained by MILP and the algorithm in the super small-scale cases (2 × 2_a), which proved the accuracy of the MILP model. Besides, four groups of experiments were carried out to demonstrate the effectiveness of HAIG. The experiment results led to the following three conclusions:

(1) Intermingling sub-lots of one lot with another effectively reduces the maximum completion time at the cost of a little increase in the sequence-dependent setup times and hence promotes production efficiency.

(2) Three modifications in HAIG, including heuristic-based initialization, adaptive local search, and improved acceptance criterion, are complementary, and hence the combination of them is effective.

(3) HAIG is significantly superior to five state-of-the-art algorithms in terms of near optimality and robustness.

Future research will consider a practical lot-streaming assembly flow shop scheduling problem in which parts are manufactured at the previous stage and are assembled into the final product at the final stage. Limited by economy production and transportation lots, all the parts are manufactured and transported in variable and intermingled sub-lots to shorten the production cycle time and reduce the work-in-process inventory. Besides, an adaptive strategy based on reinforced learning will be adopted in the destruction–construction phase to improve the performance of the algorithm.

## Figures and Tables

**Figure 1 sensors-23-02808-f001:**
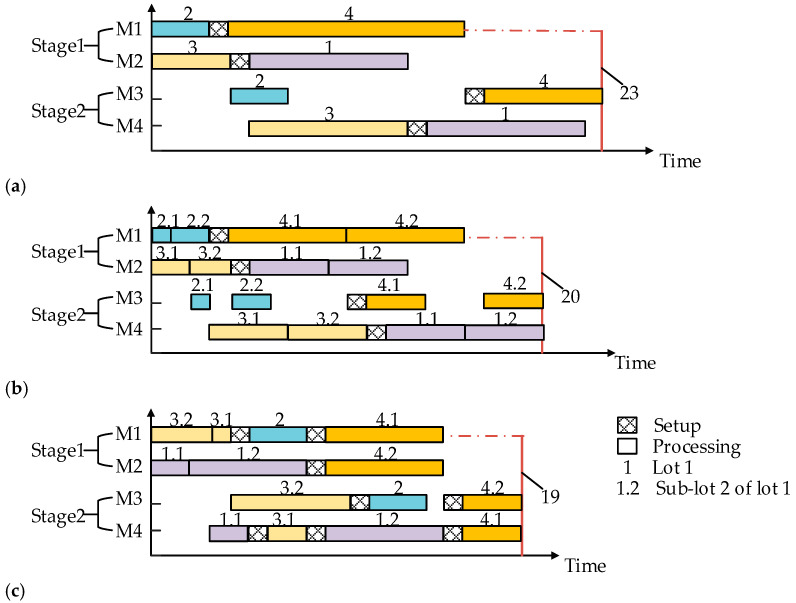
Lot-streaming HFSP with/without intermingling. (**a**) HFSP without lot-streaming; (**b**) Lot-streaming HFSP without intermingling; (**c**) Lot-streaming HFSP with intermingling.

**Figure 2 sensors-23-02808-f002:**
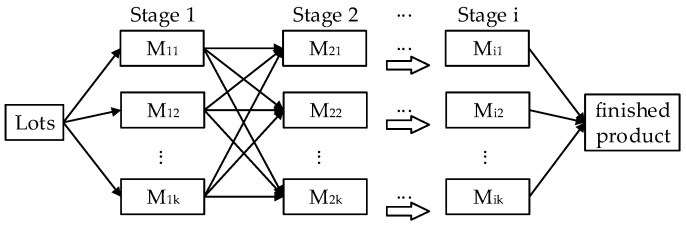
Framework of the hybrid flow shop environment.

**Figure 3 sensors-23-02808-f003:**
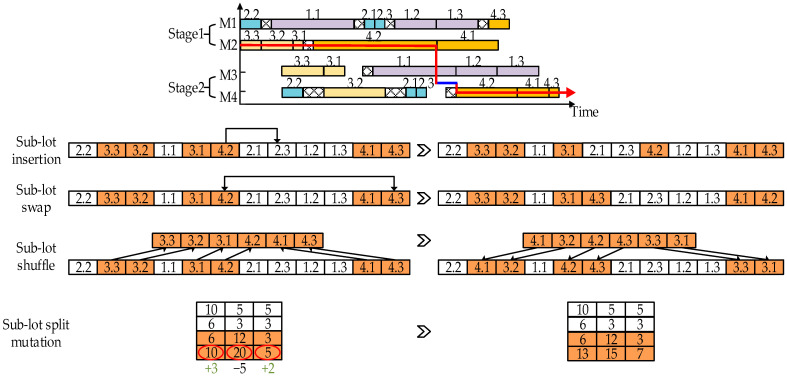
The illustrations of the neighborhood structures.

**Figure 4 sensors-23-02808-f004:**
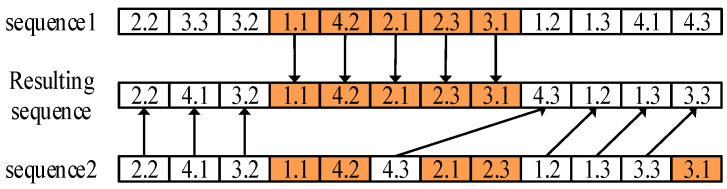
Order crossover operator for the production sequence.

**Figure 5 sensors-23-02808-f005:**
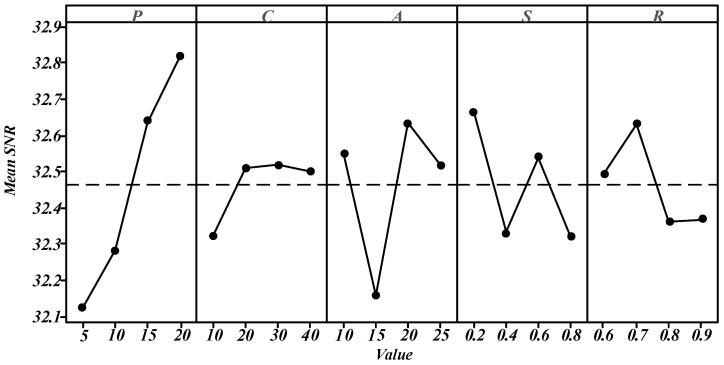
SNR main effects plot.

**Figure 6 sensors-23-02808-f006:**
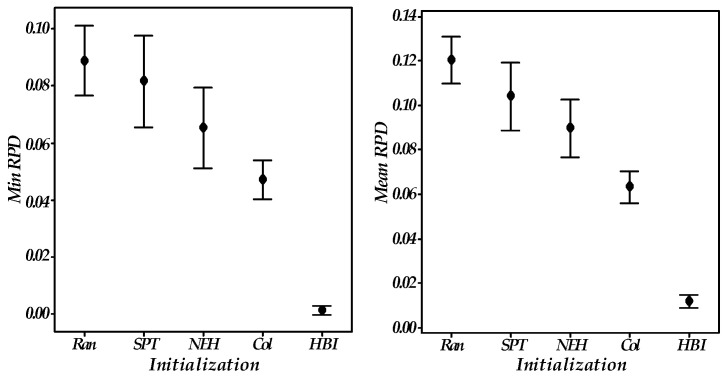
Comparisons of initialization rules.

**Figure 7 sensors-23-02808-f007:**
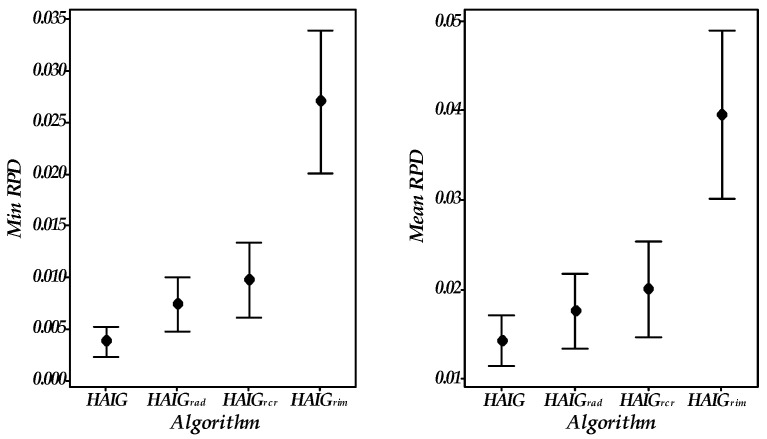
Comparisons of ${\mathrm{HAIG}}_{\mathrm{rcr}},{\;\mathrm{HAIG}}_{\mathrm{rad},}\;{\mathrm{HAIG}}_{\mathrm{rim}}$ and HAIG algorithms.

**Figure 8 sensors-23-02808-f008:**
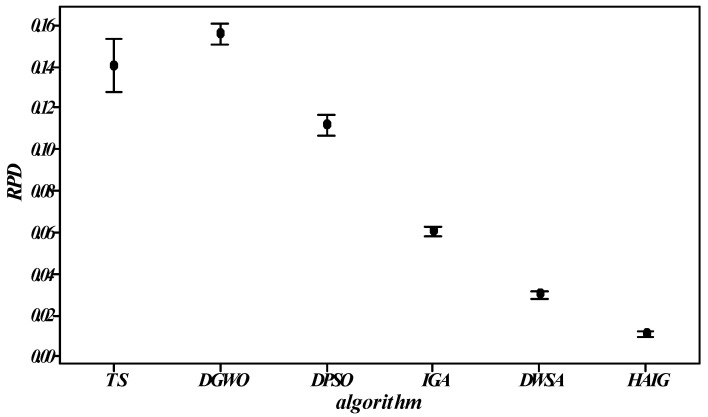
Comparisons of the algorithms.

**Figure 9 sensors-23-02808-f009:**
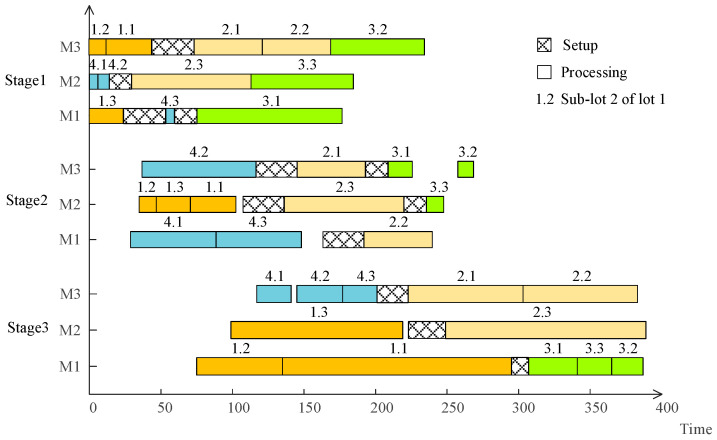
A scheduled Gantt chart for the case.

**Table 1 sensors-23-02808-t001:** The notations.

**Parameters and Sets:**	
L	The maximum number of sub-lots from each lot.
*f*	The number of stages
*v*	The number of lots
*r*	The number of machines
I	Set of stages and $I=\left\{1,\ldots i\ldots f\right\}$.
J	Set of lots and $J=\left\{1,\ldots j\ldots v\right\}$.
E	Sets of sub-lots, and $E=\left\{1,\ldots e\ldots L\right\}$.
M	Set of machines and $M=\left\{1,\ldots m\ldots r\right\}$.
$M_{i}$	Set of machines at stage $i,\;M_{i}\in M$.
$n_{i}$	The quantity of the items in lot j.
$p_{i,k,j}$	Processing time per item of lot j by machine k at stage i.
$s_{i,j,j\prime }$	Setup time between lots j and j′ on a machine at stage i.
$tr_{i}$	Transportation time from stage i to the next stage.
Ψ	A positive large number.
**Discrete variables:**	
$Q_{j,e}$	Integer variable, the quantity of the items in sub-lot e from lot j.
$W_{j,e}$	Binary variable. It takes the value of 1 when the quantity of items in sub-lot e from lot j is larger than 1 and 0 otherwise.
$D_{i,k,j,e}$	Binary variable. It takes the value of 1 when sub-lot e of lot j is allocated to machine k at stage i and 0 otherwise.
$Y_{i,k,j,e,j\prime ,e\prime }$	Binary variable. It takes the value of 1 when on machine k at stage i sub-lot e of lot j is performed immediately before sub-lot e′ from lot j′ and 0 otherwise.
**Continuous variables:**	
$B_{i,j,e}$	Beginning time of sub-lot e of lot j at stage *i*.
$F_{i,j,e}$	Finishing time of sub-lot e of lot *j* at stage *i*.
$C_{\max}$	Maximum completion time.

**Table 2 sensors-23-02808-t002:** Parameters values of other algorithms.

Algorithm	Parameter
DGWO	$\mathrm{Popsize}=20,{\;\mathrm{Local}}_{\mathrm{Max}}=3,\;\mathrm{BillLength}=7$
DPSO	$\mathrm{Popsize}=15,{\;\mathrm{Local}}_{\mathrm{Max}}=15$
IGA	Popsize=25,restart=15,Crossover rate=0.2,Mutation rate=0.8
TS	Popsize=15, BillLength=20
DWSA	$\mathrm{Popsize}=20,{\mathrm{Deduplication}}_{\mathrm{Archive}}=5,\;\mathrm{Acceptance}=0.2$

**Table 3 sensors-23-02808-t003:** Comparisons of the initialization rules.

Problem	Min RPD					Mean RPD				
	Ran	SPT	NEH	Col	HBI	Ran	SPT	NEH	Col	HBI
4×3_a	0.0325	0.04	0.018	0.005	0	0.092	0.095	0.074	0.014	0.023
4×3_b	0.0347	0.003	0.034	0.032	0	0.079	0.014	0.153	0.088	0.004
4×5_a	0.0053	0.009	0	0.002	0.002	0.067	0.032	0.045	0.028	0.02
4×5_b	0.038	0.167	0.005	0.014	0	0.104	0.167	0.12	0.055	0.004
12×3_a	0.06	0	0.018	0.053	0.026	0.087	0.025	0.069	0.104	0.039
12×3_b	0.1	0.121	0.128	0.057	0	0.157	0.155	0.142	0.064	0.022
12×5_a	0.087	0.003	0.012	0.037	0	0.144	0.038	0.03	0.092	0.01
12×5_b	0.136	0.083	0.089	0.087	0	0.175	0.112	0.138	0.114	0.012
20×3_a	0.059	0.025	0	0.037	0.004	0.095	0.046	0.036	0.06	0.019
20×3_b	0.123	0.12	0.115	0.042	0	0.145	0.16	0.13	0.055	0.008
20×5_a	0.091	0.043	0.024	0.019	0	0.141	0.107	0.043	0.082	0.012
20×5_b	0.126	0.12	0.098	0.08	0	0.148	0.15	0.123	0.093	0.014
36×3_a	0.038	0.024	0.027	0.029	0	0.077	0.044	0.045	0.045	0.016
36×3_b	0.107	0.142	0.09	0.055	0	0.146	0.162	0.106	0.057	0.006
36×5_a	0.067	0.053	0.017	0.069	0	0.115	0.073	0.026	0.093	0.027
36×5_b	0.137	0.119	0.089	0.071	0	0.159	0.14	0.115	0.075	0.006
46×3_a	0.048	0.036	0.025	0.04	0	0.07	0.061	0.036	0.05	0.014
46×3_b	0.121	0.149	0.107	0.06	0	0.151	0.17	0.119	0.06	0.004
46×5_a	0.056	0.058	0	0.027	0.015	0.089	0.072	0.018	0.044	0.027
46×5_b	0.122	0.114	0.109	0.067	0	0.15	0.133	0.127	0.067	0.008
56×3_a	0.096	0.096	0.074	0.064	0	0.125	0.114	0.088	0.08	0.006
56×3_b	0.105	0.122	0.089	0.038	0	0.13	0.143	0.099	0.04	0.003
56×5_a	0.08	0.075	0.052	0.037	0	0.114	0.104	0.062	0.058	0.012
56×5_b	0.115	0.119	0.096	0.039	0	0.133	0.13	0.104	0.044	0.007
70×3_a	0.077	0.06	0.073	0.049	0	0.08818	0.069	0.082	0.061	0.013
70×3_b	0.121	0.129	0.106	0.052	0	0.147	0.152	0.124	0.053	0.002
70×5_a	0.052	0.042	0.035	0.037	0	0.079	0.06	0.042	0.049	0.018
70×5_b	0.122	0.123	0.111	0.051	0	0.148	0.135	0.12	0.051	0.003
85×3_a	0.08	0.0576	0.069	0.07	0	0.099	0.081	0.08	0.078	0.007
85×3_b	0.124	0.139	0.115	0.053	0	0.145	0.155	0.119	0.054	0.001
85×5_a	0.058	0.051	0.048	0.04	0	0.081	0.08	0.069	0.055	0.017
85×5_b	0.139	0.112	0.11	0.054	0	0.148	0.124	0.124	0.058	0.003
100×3_a	0.092	0.09	0.093	0.081	0	0.109	0.101	0.107	0.089	0.012
100×3_b	0.132	0.12	0.116	0.056	0	0.151	0.138	0.126	0.056	0.003
100×5_a	0.084	0.051	0.051	0.042	0	0.094	0.072	0.07	0.059	0.019
100×5_b	0.135	0.127	0.112	0.053	0	0.148	0.133	0.122	0.053	0.001

**Table 4 sensors-23-02808-t004:** Comparisons of the algorithms.

Problem	TS		DGWO		DPSO		IGA		DWSA		HAIG	
	Best	Avg	Best	Avg	Best	Avg	Best	Avg	Best	Avg	Best	Avg
4×3_a	452	535.1	437	468	447	473	416	423.6	416	420.6	400	409.2
4×3_b	711	748.1	664	726.9	701	728.6	660	665.1	644	658.2	634	636.4
4×5_a	640	852.1	641	671.8	631	642.1	589	603.7	571	576.6	567	577.6
4×5_b	845	1038.4	811	902.8	955	963.2	810	813.3	794	804.3	760	763
12×3_a	1051	1112.9	1098	1122.3	1054	1085.1	1023	1049.7	1030	1037.5	1004	1017.2
12×3_b	2157	2214.5	2300	2377.2	2204	2234.1	2045	2067.5	1975	2021.1	1928	1970.6
12×5_a	1325	1502.5	1315	1374.1	1211	1243.8	1183	1235.3	1173	1188	1167	1178.5
12×5_b	2229	2389.7	2443	2554.2	2320	2383.5	2180	2264.1	2138	2170.5	2054	2078
20×3_a	1714	1855.6	1741	1802.1	1673	1712.6	1648	1674.7	1642	1671.1	1609	1632.4
20×3_b	3932	4106.1	4383	4470.7	4236	4318.6	3935	3981.6	3849	3879.8	3753	3782.8
20×5_a	2026	2303.3	1949	2083.9	1910	1922.3	1886	1971.8	1811	1847.5	1789	1811
20×5_b	4159	4221.5	4501	4601.6	4364	4473	4134	4165.1	4010	4066.6	3874	3930.1
36×3_a	2923	3022.1	2953	3008.9	2885	2915.1	2821	2864.6	2839	2860.4	2736	2779.5
36×3_b	7234	7305.3	7802	7912.8	7237	7291.7	6968	7005.1	6690	6701.6	6586	6627.1
36×5_a	3268	3510.8	3327	3429.2	3128	3172.1	3233	3319.8	3145	3185.1	3047	3129.6
36×5_b	7140	7328.7	7824	8003.5	7561	7640.1	7110	7190.7	6784	6883.9	6696	6738.7
46×3_a	3996	4058	3868	3960	3810	3846	3743	3791	3735	3790	3633	3684.9
46×3_b	9683	9861.2	10,412	10,570.2	9820	9895.8	9326	9361.2	8925	8939.4	8811	8844.4
46×5_a	4091	4339.9	4292	4332.3	4066	4094.2	4013	4053.9	4011	4056.5	3996	4043
46×5_b	9504	9696.4	10,463	10,615.2	9994	10,123.3	9507	9536.4	9131	9158	8933	9000.1
56×3_a	4910	5112.9	5157	5235.7	4906	4993.2	4787	4856.6	4718	4769.4	4563	4590.3
56×3_b	13,783	13,894.1	14,823	14,936	13,932	14,050.6	13,281	13,330.6	12,913	12,924.1	12,768	12,810.8
56×5_a	5289	5455.8	5385	5538.3	5233	5268.9	5056	5130.6	5046	5077	4919	4977.5
56×5_b	13,673	13,877.1	14,899	14,987	14,241	14,335	13,497	13,538.9	13,013	13,151.7	12,902	12,998.3
70×3_a	5797	5949.7	6028	6154.3	6039	6112.7	5879	5917.5	5751	5808.6	5592	5666.2
70×3_b	16,159	16,440.5	17,489	17,678.5	16,722	16,862	15,679	15,694.9	14,932	15,016.7	14,887	14,913.7
70×5_a	6283	6428.4	6378	6528.1	6276	6323.1	6194	6301.3	6014	6109	6003	6112.8
70×5_b	16,235	16,632.6	17,399	17,603.4	16,729	16,982.2	15,853	15,878	15,203	15,247.8	15,038	15,080
85×3_a	7481	7746.1	7738	7872.6	7725	7765.2	7536	7639.7	7264	7302	7118	7166.7
85×3_b	21,266	21,692.5	22,867	23,039	21,824	21,953.9	20,624	20,661.6	19,641	19,650.8	19,543	19,571
85×5_a	7744	7941.9	8035	8241.3	8019	8073.8	7792	7884.3	7645	7720	7505	7631.1
85×5_b	21,320	21,646.7	22,717	22,931	22,107	22,247	20,776	20,880.8	19,869	19,879.4	19,699	19,756.7
100×3_a	8626	8841.8	8962	9091.3	8929	9012.6	8669	8756.2	8398	8437	8097	8191
100×3_b	24,610	25,354.8	26,619	26,776.3	25,513	25,597.9	23,816	23,817.8	22,648	22,678	22,522	22,583
100×5_a	8929	9009.1	9278	9414.2	9323	9438.1	8881	8984.5	8827	8924.1	8658	8824.6
100×5_b	24,622	24,974.2	26,417	26,619	25,630	25,722.4	23,935	23,972	22,875	22,914.6	22,712	22,738.9

**Table 5 sensors-23-02808-t005:** The case data. (**a**). Production data for the illustrative example. (**b**). Production data for the illustrative example.

**(a)**								
**Lot**	**Stage 1**	**Stage 2**	**Stage 3**	**Unit**	**Sequence-Dependent Setup Time/Stages**			
					**Lot1**	**Lot2**	**Lot3**	**Lot4**
lot1	A1	B1	RF9	34	0, 0, 0	30, 30, 26	30, 30, 12	30, 19, 21
lot2	A2	B2	WB4	30	30, 30, 26	0, 0, 0	0, 16, 19	16, 29, 22
lot3	A2	B3	WB1	40	30, 30, 12	0, 16, 19	0, 0, 0	16, 21, 21
lot4	A3	B4	WB5	20	30, 19, 21	16, 29, 22	16, 21, 21	0, 0, 0
**(b)**								
**Stage**				**Transportation Time**	**Unit Time per Item/Lots**			
Stage 1				23	2, 6, 6, 1			
Stage 2				28	2, 6, 1, 10			
Stage 3				-	10, 10, 2, 4			

## Data Availability

Not applicable.

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
