# Peer review of "A Heuristic-Based Adaptive Iterated Greedy Algorithm for Lot-Streaming Hybrid Flow Shop Scheduling Problem with Consistent and Intermingled Sub-Lots"

_sensors, 2023, doi:10.3390/s23052808_

Round 1

Reviewer 1 Report

This paper studies the lot-streaming hybrid flow shop scheduling problem with consistent and intermingled sub-lots (LHFSP-CIS). An MILP model and a heuristic-based adaptive iterated greedy algorithm are designed. This paper is very interesting and well written. However, some minor revisions must be finished before final acception.

(1) The literature review is not adequate. The related papers that are focused on exact methods (mainly MILP models) for solving scheduling problems should be reviewed.

(2) The results of MILP model should be compared with HAIG in Section 5.

Author Response

Thank you for your decision and constructive comments on my manuscript. We have carefully considered the suggestion of Reviewer and make some changes. We have tried our best to improve and made some changes in the manuscript. Here we simply described our revision.

1 “The literature review is not adequate. The related papers that are focused on exact methods (mainly MILP models) for solving scheduling problems should be reviewed.

 The exact methods were modified in the abstract.

2 “The results of MILP model should be compared with HAIG in Section 5.

The result of MILP model compared with HAIG is show in the conclusion.

Reviewer 2 Report

In the manuscript considered a several optimization methods for lot-streaming hybrid flow shop scheduling problem with consistent and intermingled sub-lots. A mixed integer linear programming and heuristic-based adaptive iterated greedy algorithm are proposed. Next, algorithms are compared with the state of the art methods. In my opinion, the article is suitable for publication after minor corrections. My main comments are as follows:

In Literature review a small paragraph about setups is missing.

Notation of set and his size is to similar. It will be better to use a calligraphy font for sets.

Usually the setup time is denoted by s.

Form of equation (1) is untype, authors add new variable/parameters Z and this is its only use.

L_j is The maximum number of sub-lots for j lot?

In equation (6) is sum from k=0 to M_i, where M_i is a set. In algorithm 2 (line 4) is similar mistake. 

In algorithm 3 (line 11) a M_1 is correct? 

In experiments for TS, IGA, DPSO, DWSA and DGWO, the full name of the algorithms should be given on first appearance.

On Figures 4, 5, 6, 7 the axis name X is missing.

Author Response

Thank you for your decision and constructive comments on my manuscript. We have carefully considered the suggestion of Reviewer and make some changes. We have tried our best to improve and made some changes in the manuscript. Here we simply described our revision.

1 “In Literature review a small paragraph about setups is missing.”

The information about the setup was added to the literature review(page 3). 

2 “Notation of set and his size is to similar. It will be better to use a calligraphy font for sets.”

3 “Usually the setup time is denoted by s.”

4 “Form of equation (1) is untype, authors add new variable/parameters Z and this is its only use.”

The notations of set and setup time were modified and were replaced in later text and pseudocode. Besides, the variable/parameter Z was deleted.

5  is The maximum number of sub-lots for j lot?”

6 “In equation (6) is sum from k=0 to , where  is a set. In algorithm 2 (line 4) is similar mistake.”

 I'm sorry I didn't express myself clearly. Actually, there is not an .The maximum number of sub-lots for each lot, L, is a fixed value, it is not a set.

The expression is true for any stage in the set I, iI; The  in equation (6) and Algorithm 2 assign a value to i before using.

7 “In algorithm 3 (line 11) a  is correct?”

 After our careful inspection, is correct, initialization is optimized for the production sequence and sub-lot splitting in the encode, while the machine sequence in the code is only used for the first stage of machine allocation.

 8 “In experiments for TS, IGA, DPSO, DWSA and DGWO, the full name of the algorithms should be given on first appearance.”

  The full names of the algorithms were given on page 15.

9 “On Figures 4, 5, 6, 7 the axis name X is missing.”

The axis names X of Figures 4, 5, 6, and 7 were added on page 15,17,18,and 20.

Reviewer 3 Report

- The abstract should be rewritten. Some numerical results should be included in the abstract. 

- A block diagram of the HFS environment and the case study problem may be included. 

- None discrete event simulation framework (or similar) are described, nor introduced.

There are some typos in the full paper. The writing of the manuscript can be further improved to reach the journal quality.

- The limitations of the proposed algorithm should be explained. 

The future works need to be presented in detail.

Author Response

Thank you for your decision and constructive comments on my manuscript. We have carefully considered the suggestion of Reviewer and make some changes. We have tried our best to improve and made some changes in the manuscript. Here we simply described our revision.

1 “The abstract should be rewritten. Some numerical results should be included in the abstract.”

The Non-parametric Kruskal-Wallis test is added to assess the effectiveness of the algorithm, and the test value (p=0) shows significantly superior.

2 “A block diagram of the HFS environment and the case study problem may be included.”

The block diagram of the HFS environment is added to the problem formulation (page 5). And the case study is in section 5.6(page 20).

3 “None discrete event simulation framework (or similar) are described, nor introduced.

Discrete event simulation frameworks are on the introduction (page 2) and the industrial case (page 20) owing to the Gantt chart is a kind of Discrete event simulation framework.

4 “There are some typos in the full paper. The writing of the manuscript can be further improved to reach the journal quality.

We apologize for the poor language of our manuscript. We worked on the manuscript for a long time and the repeated addition and removal of sentences and sections obviously led to poor readability. We have now worked on both language and readability. We really hope that the flow and language level have been substantially improved.

5 “The limitations of the proposed algorithm should be explained.”

The limitation of the proposed algorithm that the proposed algorithm is only validated for this problem and cannot guarantee its performance for other problems is explained in the Effectiveness of the proposed HAIG (page 18).

6 “The future works need to be presented in detail.”

Future work is re-elaborated and added about the algorithm.